# A Mental Workload Classification Method Based on GCN Modified by Squeeze-and-Excitation Residual

**Zheng Zhang [1], Zitong Zhao [1], Hongquan Qu [1,*], Chang'an Liu [1] and Liping Pang [2]**

[1] School of Information Science and Technology, North China University of Technology, Beijing 100144, China
[2] School of Aeronautic Science and Engineering, Beihang University, Beijing 100191, China
*   Correspondence: qhqphd@ncut.edu.cn

**Abstract:** In some complex labor production and human–machine interactions, such as subway driving, to ensure both the efficient and rapid completion of work and the personal safety of staff and the integrity of operating equipment, the level of mental workload (MW) of operators is monitored at all times. In existing machine learning-based MW classification methods, the association information between neurons in different regions is almost not considered. To solve the above problem, a graph convolution network based on the squeeze-and-excitation (SE) block is proposed. For a raw electroencephalogram (EEG) signal, the principal component analysis (PCA) dimensionality reduction operation is carried out. After that, combined with the spatial distribution between brain electrodes, the dimensionality reduction data can be converted to graph structure data, carrying association information between neurons in different regions. In addition, we use graph convolution neural network (GCN) modified by SE residual to obtain final classification results. Here, to adaptively recalibrate channel-wise feature responses by explicitly modelling interdependencies between channels, the SE block is introduced. The residual connection can ease the training of networks. To discuss the performance of the proposed method, we carry out some experiments using the raw EEG signals of 10 healthy subjects, which are collected using the MATB-II platform based on multi-task aerial context manipulation. From the experiment results, the structural reasonableness and the performance superiority of the proposed method are verified. In short, the proposed GCN modified by the SE residual method is a workable plan of mental workload classification.

**Keywords:** mental workload; graph convolution network (GCN); squeeze-and-excitation; residual connection

**MSC:** 68T01

## 1. Introduction

In recent years, brain–computer interfaces (BCIs) [1,2] have been used as devices for connecting the human body to external devices, serving for biological research, disease treatment, etc. Here, due to the non-invasive nature of BCIs, electroencephalogram (EEG) signals are usually collected based on it, a method which is widely used. Examples include emotion recognition [3], sleep stage assessment [4], and mental workload (MW) classification [5]. Among them, MW represents the amount of brain activity per unit of time. The principal reason for measuring workload is to quantify the mental cost of performing tasks in order to predict operator and system performance. If the MW is too high, it can lead to brain fatigue and reduced levels of alertness, which can easily lead to decision errors [6,7]. In contrast, low MW can lead to negative burnout, which to some extent causes a waste of human resources [8,9]. Therefore, it is important to assess the MW of operators in real time to help rationalize the use of resources and improve work efficiency [10,11]. The continuous and objective measurement of a few physiological indicators of the operator also enables the assessment of MW. Examples include EEG signals, eye movements, heart

rate, and respiration. Although the EEG signal is complex, it visually reflects the electrical activity of the brain. Moreover, EEGs contain the most valid information compared to other human bioelectrical signals. Thus, in this paper, we focus on the MW classification task with EEG.

As indicated, there are three types of MW classification methods: subjective perception of the subject, job performance, and physiological indicators related to MW [12]. Among the methods based on the subjective perceptions of subjects are the subjective workload assessment technique (SWAT), the NASA task load index (NASA-TLX), and the workload profile (WP) [13]. Performance measurement is to measure the operating performance of the subject based on tasks [8,14]. These three types of methods are susceptible to human factors, causing a lack of objectivity and relevance. Moreover, real-time monitoring cannot be realized by these methods, so practical requirements cannot be satisfied. Therefore, MW classification methods need to be further studied [15–17].

During the development of machine learning [18], many methods have been applied to the EEG MW classification task, for example, the support vector machine (SVM) [19] employed by Qu et al. [20] and transfer learning utilized by Zheng et al. [21]. In particular, there has been focus on the power spectral density (PSD) of raw EEG signals, and on this basis, PSD are used to output EEG cloud maps before feeding into the network for feature extraction. Kose et al. [22] used a weighted functional brain network (WFBN) to analyze dynamic MW conditions. In addition to this, the dynamic changes in the topology of brain connectivity for MW conditions are evaluated and characterized using an advanced technique developed based on the weighted edges ordinal sequence (WEOS). Finally, k nearest neighbor (KNN), SVM, and random forest (RF) are used to complete the classification.

With the growth of deep learning, numerous models with powerful feature extraction capabilities are used for MW classification. Pang et al. use a stochastic configuration network (SCN) and subject-specific classifiers (SSCs) to achieve good classification results. Umer Asgher [23] et al. accomplish MW classification using SVM and a convolutional neural network (CNN), respectively. The experimental results show that the tuned CNN network is stronger than SVM for the datasets used in the paper. Debashis et al. [24] take full advantage of the correlation between continuous signals in the time series. They propose a deep hybrid model based on bidirectional long short-term memory (BLSTM) and long short-term memory (LSTM) for the classification of MW. To address cross-subject classification, Yu et al. [25] propose a capsule network capturing the structural relationship between power spectral density and brain connectivity features.

These methods have three drawbacks. Firstly, the traditional machine learning methods of EEG signal analysis do not take full advantage of the rich spatial information contained in the EEG signal and the correlations of neurons in different regions. For example, the default sorting is directly performed on the data nodes of a regular Euclidean space such as the sample object, meaning that the deep association information cannot be extracted. That is, the spatial structure between nodes is ignored. Next, in the fitting process of subsequent deep learning models, such as CNN and LSTM, which are able to learn the connection of each node spontaneously, the extraction of correlation between nodes is not sufficiently in-depth because of factors such as an insufficient number of training rounds and multiple noises. Finally, the popular EEG signal analysis perspectives are raw signal and frequency domain signal. In the conversion to frequency domain after pre-processing of the raw EEG signal, feature loss is also inevitable in the mapping using features such as PSD. Further, in this process of secondary processing of data, the amount of data will be continuously reduced, which may result in the inevitable loss of high-level features [21]. Thus, the subsequent training of the model is also prone to overfitting due to the small amount of feature selection.

In our method, the use of pre-processed raw EEG signals as the input of the graph convolutional network (GCN) can theoretically solve these two existing drawbacks. In terms of data, principal component analysis (PCA) [26] feature selection is performed

after raw data pre-processing. In order to eliminate the "edge" electrodes that have little impact on the results, the data are downscaled before being sent to the model training. Meanwhile, reducing the amount of data in advance can also alleviate the problem of excessive system memory consumption in the subsequent graph convolution training to a certain extent. In terms of the model, the proposed one refers to the squeeze-and-excitation (SE) structure and combines residual connection based on the GCN [27,28]. The GCN network employed is ChebNet [29] and the convolution kernel is a Chebyshev polynomial. Every three convolutional layers form one residual block. The model has four residual blocks and eight convolutional layers [30]. The embedded residual connection solves the gradient disappearance and explosion caused by the increasing number of layers in the network. Finally, an attention mechanism, namely SE block, is added after each layer of the network [31]. In order to select and focus on the most important electrode features, different weights are assigned to each EEG electrode after convolution, pooling, and full connection. The classification ability of the proposed method is strong.

## 2. Materials and Methods

### 2.1. Overall Structure of the Proposed Method

The overall model is divided into two parts: the hand-crafted feature extraction and network model.

In the hand-crafted feature extraction part, firstly, PCA dimensionality reduction is used to exclude individual useless dimensional information. Secondly, the EEG signals collected from multiple nodes are used to shape multiple node topographies containing node spatial structure information using Pearson correlation coefficients. Then, they are fed into the model in turn.

The network model consists of the ChebNet convolution, residual connection, and SE block. The data are sequentially processed using graph convolution, activation, normalization process, and pooling. Finally, there is the full link layer. MW can be divided into two categories, high MW (HMW) and low MW (LMW). HMW and LMW classification is carried out using the softmax function. To better illustrate the fundamentals of the proposed model, the specific units in the network framework are described in detail in the following sections. The overall model framework of the system is shown in Figure 1.

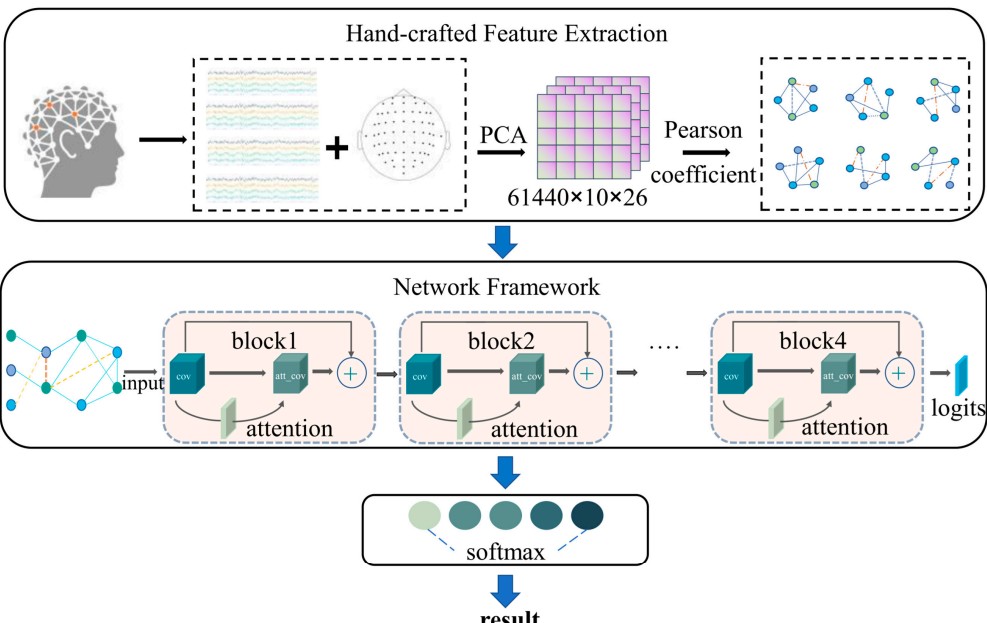

**Figure 1.** The overall structure of EEGCN.

### 2.1.1. Hand-Crafted Feature Extraction

After data collection, the temporal domain signal retains the most valid information, although it has redundancy. Moreover, the temporal domain signal can visually reflect the signal transformation with time, so we hand-crafted the extraction temporal domain features. The temporal domain features are divided into two categories: content-order and dimensionless. Firstly, the content-order features include mean, standard deviation, first-order difference, normalized first-order difference, etc. Moreover, the dimensionless features include skewness, kurtosis, waveform factor, impulse factor, etc.

The one-dimensional signals collected from multiple independent electrodes are combined with the position information between electrodes. Afterwards, PCA is used to reduce the dimensionality. The specific experimental comparison results are shown in Section 3. After that, Pearson's index between dimensions is calculated for the reduced dimensional multidimensional data. The obtained Pearson coefficients between dimensions are then used to construct a graph structure. At the end of the hand-crafted feature extraction, these graph data are fed into the network part.

### 2.1.2. Network Framework

In the network model part, the temporal domain graph features of the EEG signal are used as the input features of the graph convolution model. Firstly, they go through the ChebNet convolution operation to extract the position correlation between electrodes. Then, they go through the activation layer to improve the ability of the model to fit nonlinear problems. After that, the normalization process is used to generalize the statistical distributivity of uniform samples, which can speed up the convergence. Finally, using a pooling layer for the size reduction of features, the computational cost of the model is decreased.

The model proposed in this paper introduces the residual connection and SE block based on GCN [32,33]. Here, every two-graph convolution operation is linked by residual connection [34]. Moreover, the attention mechanism, i.e., SE [35] block, is added in each residual connection to adjust the channel-wise feature response. The structure of the SE block is shown in Figure 2. The SE block is co-trained with the network to focus on the more important electrodes in the current classification.

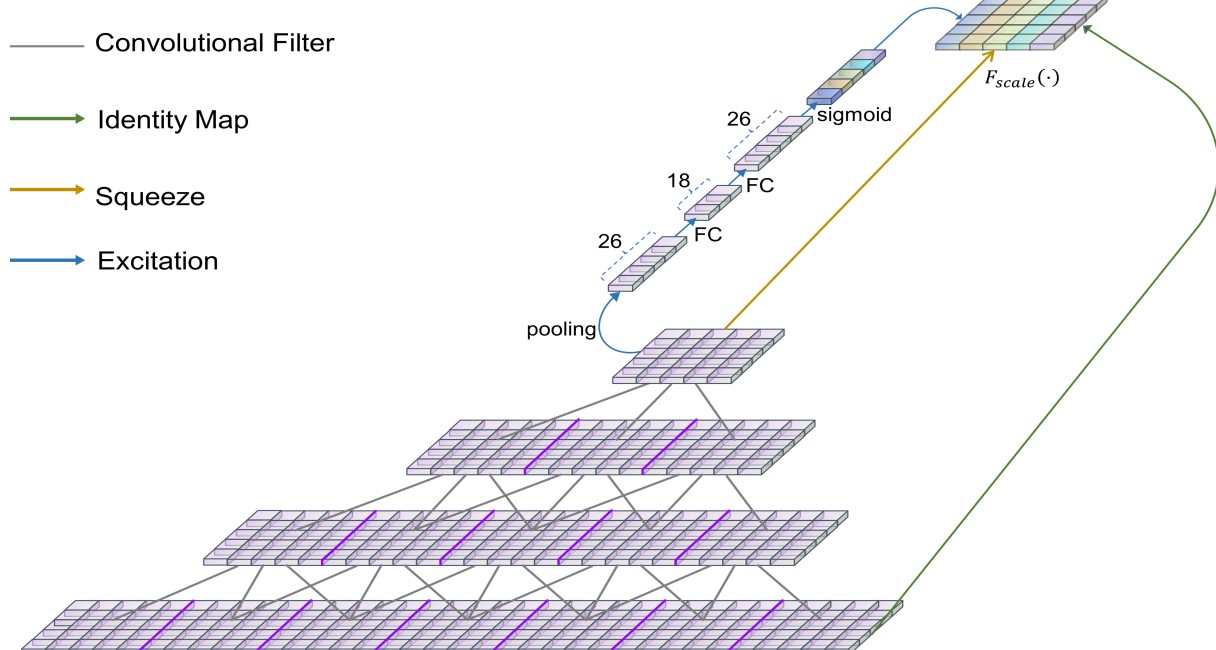

**Figure 2.** SE Block Structure.

In the SE block part of the attention mechanism, firstly, the squeeze function extracts the multi-dimensional features after convolution. Then, weights are generated for each dimension based on the parameter extracted from the excitation function, which can be seen as the importance of each dimension after feature selection. Finally, the weights are weighted to the previous features dimension by dimension using multiplication. Meanwhile, the gradient explosion is prevented by combining inter-convolutional information through the identity map operation.

### 2.2. The Key Operation of the Proposed Method
### 2.2.1. GCN

Many studies have shown that the human brain has small-world properties. There are correlations between single neurons that are consistent with a topological graph structure. Moreover, multiple independent one-dimensional signal intervals acquired by multi-wire electrodes that mimic the shape of the human brain can also be considered as graph structures [28,32]. However, when dealing with the data of graph structure type, the traditional network models ignore the information of the connection relationship between its individual nodes and cannot dig deeper into its features and patterns based on topological correlation, for example, LSTM [33] and CNN [34,35]. However, from Pearson's heat map, it can be seen that there is a correlation between the 30 brain electrodes after excluding the reference electrodes. Additionally, there is a pattern for most of the electrodes: the closer the position, the stronger the correlation, as shown in Figure 3.

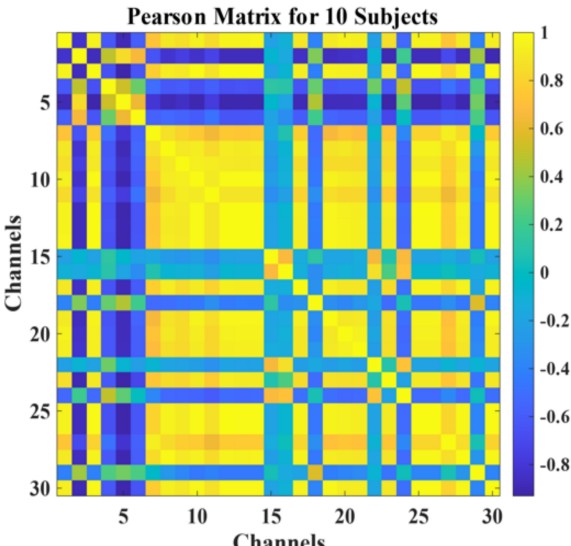

**Figure 3.** Inter−electrode Pearson's heat map.

To solve this problem, this model uses the method of spectral convolution in a graph convolution network (GCN). The Fourier transform is used to convert the graph to a Cartesian coordinate system, so that this matrix can be processed using the normal convolution principle according to the definition of convolution operation, i.e.,

$$\mathcal{F}[f_1(t) * f_2(t)] = F_1(w) \cdot F_2(w) \tag{1}$$

where $f(t)$ is the signal in the temporal domain, $F_1(w)$ is the signal in the frequency domain, $\mathcal{F}$ is the Fourier transform, $*$ denotes the convolution, and $\cdot$ denotes the product. Bruna et al. [36] first proposed the use of the Laplace matrix to accomplish the convolution

operation on the spectral domain. The Laplace matrix is defined as $L$ and normalized as: $L = I_N - D^{-1/2}AD^{-1/2}$ where $D$ is the degree matrix and

$$D_{ii} = \sum_{j=1}^{N} A_{ij} \tag{2}$$

$A$ is the adjacency matrix and $I_N$ is the unit matrix. The feature decomposition of $L$ using the graph Fourier transform obtains $L = U\Lambda U^T$, where $U$ is the eigenvector matrix of $L$ and $\Lambda$ is the eigenvalue diagonal matrix. Therefore, the graph convolution can be defined as $y = Ug_\theta(\Lambda)U^Tx$, where $g_\theta(\Lambda)$ is the Chebyshev polynomial, which is the most commonly used convolution kernel for spectral domain convolution, and the polynomial is specified as follows.

$$g_\theta(\Lambda) = \sum_{k=0}^{K-1} \theta_k T_k(\Lambda), \ \theta \in \mathbb{R}^K \tag{3}$$

$T_k(\Lambda)$ is the $k$ approximation of the Chebyshev polynomial of order, so the convolution formula can be simplified as

$$y = \sum_{k=0}^{K-1} \theta_k T_k(L)x \tag{4}$$

### 2.2.2. SE Residual Block

Many studies show that as the depth of the network increases, the network shows degenerate spectral convolution, leading to an increase in training error [37]. In order to keep mining the high-level correlation features between data in continuous convolution, a residual connection is introduced in the model. This structure is shown in Figure 4. The stacking layer occurs when the residual, F(x), is 0. When F(x) = 0, the stacking layer only performs constant mapping at this point, so at least the network performance does not degrade. Moreover, the residual connection will not actually be 0. This also allows the stacking layer to learn new features based on the input features and, thus, have better performance.

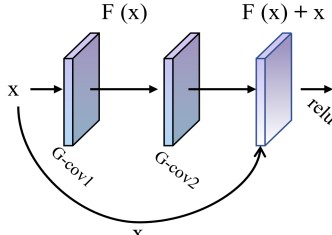

**Figure 4.** Residual connection model.

In the case of limited computational power, it is necessary to allocate computational resources to more important tasks [38,39]. A resource allocation scheme, i.e., attention mechanism, is created in neural networks. An embeddable type of attention mechanism module is proposed in the SENet network. The core operations are two blocks, squeeze and excitation, where squeeze encodes the entire spatial feature as a global feature, using global average pooling to achieve this.

$$z_c = F_{sq}(u_c) = \frac{1}{H \times W}\sum_{i=1}^{H}\sum_{j=1}^{W} u_c(i,j), z \in R^C \tag{5}$$

The excitation operation uses the information collected by the excitation operation to obtain the complete dependence between channels. This is achieved by using two fully connected layers and two activation layers. The purpose of this operation is firstly that it learns the nonlinear interactions between different features. Secondly, it focuses on all

features, instead of enforcing a single thermal activation. The excitation operation can be represented by the following equation:

$$s = F_{ex}(z, W) = \sigma(g(z, W)) = \sigma(W_2 \text{ReLU}(W_1 z)) \tag{6}$$

where $W_1 \in R^{\frac{C}{r} \times C}$ and $W_2 \in R^{C \times \frac{C}{r}}$. In order to control the computational effort while deepening the model depth, the model adds two bottleneck structure fully connected layers [39,40]. Finally, we multiply the activation values of each node learned with the original features. The flexibility of the SE block means that it can be ported to all existing networks. The proposed model convolution kernel is set to the Chebyshev convolution. The process and results of the evaluation experiments on the model in order to test the fitting ability and generalization ability of the proposed model will be described in detail in the following sections [41].

## 3. Results

### 3.1. Evaluation Criteria and Data Processing

#### 3.1.1. Data Collection

The EEG data are collected in this experiment using the MATB-II [42] platform in the Neuroscan Neuamps system. The sampling rate is 1024 Hz. The bandpass filter is set to 0.1–200 Hz. The number of channels with a time interval of 60 s is 32. Among the 32 electrodes, A1 and A2 are the reference electrodes, and the reference diagram is shown in Figure 5. The specific data collection methods will be described in detail in Section 3.

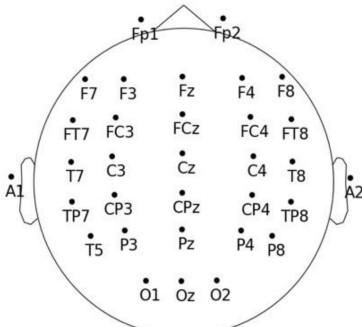

**Figure 5.** Diagram of electrode position.

Based on the MATB-II [43] platform, we design four tasks with different EEG loads to monitor the subjects' status of completing the tasks sequentially in real time [44]. The four tasks include information monitoring of system instrument scales, vehicle tracking, air traffic control communication tasks, and fuel resources. The subjects need to manipulate the mouse and flight joystick to make corresponding operations when facing unexpected situations and task transitions. The interface of the experimental task is shown in Figure 6. The system monitoring task is presented in the upper left window of the display. Here, the requirements for monitoring the instruments and warning lights are simulated, while the requirements for manual control are simulated by the tracking task. This task is displayed in the upper middle window. During the simulation, the communication task provides the operator with pre-recorded auditory messages at selected intervals, while the requirements for fuel management are simulated by the resource management task. The four areas correspond to the visual and operational tasks. The two types of loads monitored are LMW and HMW. The details of the four sub-tasks are shown in Table 1.

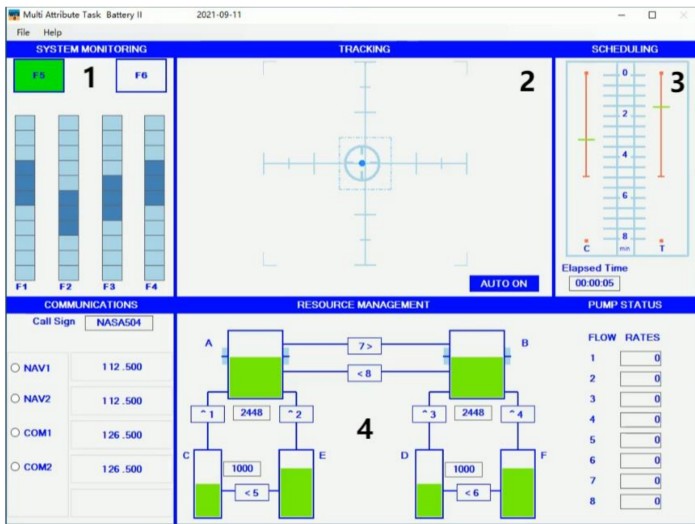

**Figure 6.** Area 1,2,3 and 4 are four tasks interface of MATB-II platform.

**Table 1.** Details of four tasks.

| Task Title | Description | Frequency | |
|---|---|---|---|
| | | LMW | HMW |
| System Monitoring | Monitor the scales of F1–F4 in Area 1 and respond with a mouse when the scales are not around the center. | | |
| Tracking | In Area 2, keep the target at the grid center by using the joystick in MANUAL mode, while no action is required in AUTO mode. | 1 | 24 |
| Scheduling | Monitor scheduling bar in Area 3 and respond to the activated communication with keyboard immediately. | | |
| Resource Management | Monitor oil volume in tanks and pump status in Area 4. Click the corresponding oil pump with the mouse when a failure occurs. | | |

The ten subjects selected for this experiment are all graduate students with an engineering background from the Beijing University of Aeronautics and Astronautics, aged 22–25 years old. All of them are male and in good health. Before the start of the experiment, they are trained to understand the overall experimental procedure and the corresponding operations to be undertaken for different tasks to avoid experimental errors caused by unskilled operations.

3.1.2. Classification Evaluation Metrics

In order to comprehensively evaluate the performance of the model, three metrics are used. These include *Kappa*, *F*1 index, and Acc. The *Kappa* metric is defined as:

$$kappa = \frac{p_0 - p_e}{1 - p_e} \tag{7}$$

Here, $p_0$ is the classification accuracy and $p_e$ is the probability of expected and actual agreement. The *Kappa* index is used to indicate the degree of classification recognition agreement. The closer the index is to 1, the stronger its consistency is.

The *F1* metric is defined as:

$$F_1 = \frac{2\,\text{Precision} * \text{Recall}}{\text{Precision} + \text{Recall}} \tag{8}$$

Here, the Precision and Recall metric are defined as:

$$\text{Precision} = \frac{TP}{TP + FP} \tag{9}$$

$$\text{Recall} = \frac{TP}{TP + FN} \tag{10}$$

All the experimental data can be classified into the following four cases: true positive (*TP*), false positive (*FP*), true negative (*TN*), and false negative (*FN*).

### 3.1.3. Data Processing

PCA [26,45] is a multivariate statistical method to examine the correlation between multiple dimensions [46]. In the proposed model, the 30-dimensional raw EEG signals extracted from multiple pathways are subjected to dimensionality reduction, which is necessary to mitigate the data redundancy. Therefore, in order to compare the classification effects of dimensionality reduction to different dimensions and find the optimal dimensionality reduction scheme, the 30-dimensional data are processed to 29, 28, 27, 26, 25, and 24 dimensions and fed into the SVM, respectively. The average accuracy comparison of the model classification is shown in Figure 7. The results after several rounds of experiments show that the 24-dimensional effect is the worst, with 83.5%. The 26-dimensional results are slightly better than the 27- and 25-dimensional. Therefore, after dimensionality reduction, the 26-dimensional data are fed into the proposed model.

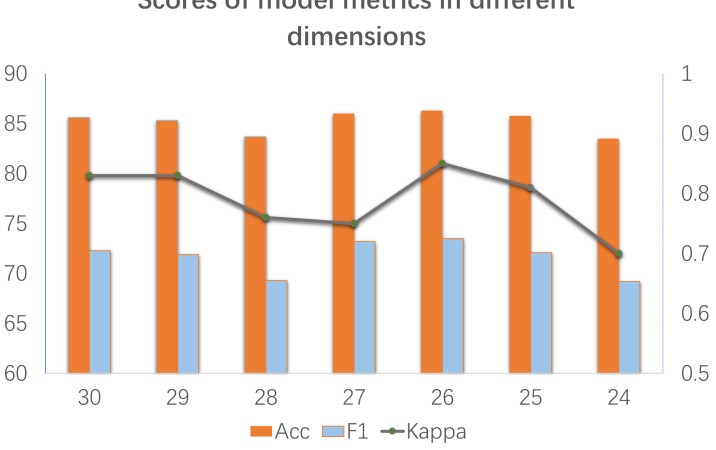

**Figure 7.** Scores of model metrics in different dimensions.

### 3.2. Experimental Settings

In this experiment, HMW and LMW are collected separately for each subject. The experiment is performed twice a day for a total of 10 days. The dataset is cut into training and test sets according to the ratio of 9:1. The data obtained are cut, with a cut criterion of 30 s at a time. Further, the number of samples obtained for each subject at a time is 30,720, with a total number of samples for 10 subjects of $10 \times 30{,}720 \times 18 \times 2$.

Next, we introduce the hyper-parameter settings of the model. The hyper-parameter settings are all adjusted after continuous trial and error to best suit the present model. The batch size is set to 1024 before the model starts training for a total of 200 epochs. The Chebyshev polynomial with the highest term count of 3 is used as the convolution kernel of the network. The number of convolution kernels is set to 16, 32, 64, and 128 for each of the four block layers in order. To prevent overfitting, the random deactivation rate is set to

0.5. The learning rate is initially set to $1 \times 10^{-6}$, and the decay rate is set to 1. The learning rate is updated afterwards using the exponential decay method. The adaptive moment estimation (Adam) optimizer is used for parameter learning, and the model is run on an NVIDIA 2080Ti GPU using the software framework Tensorflow 1.14.1.

### 3.3. Proposed Network Evaluation

In order to evaluate the classification ability of the proposed model at different stages, t-SEN charts are firstly drawn at different steps, as shown in Figure 8. The data of subject 1 are plotted as an example. The three states of data presented are raw data, Block2, and Softmax layers. From Figure 8, we can observe that the data distribution is very chaotic and poorly differentiable before the data go into the model. After the hand-crafted feature extraction and the two-layer block structure, the distribution of the two types of data in 2D becomes gradually clear. The last layer shows that the HMW and LMW are roughly separated in two regions, although there are still errors. It can be concluded from this that the model proposed in the paper allows an excellent classification in terms of MW.

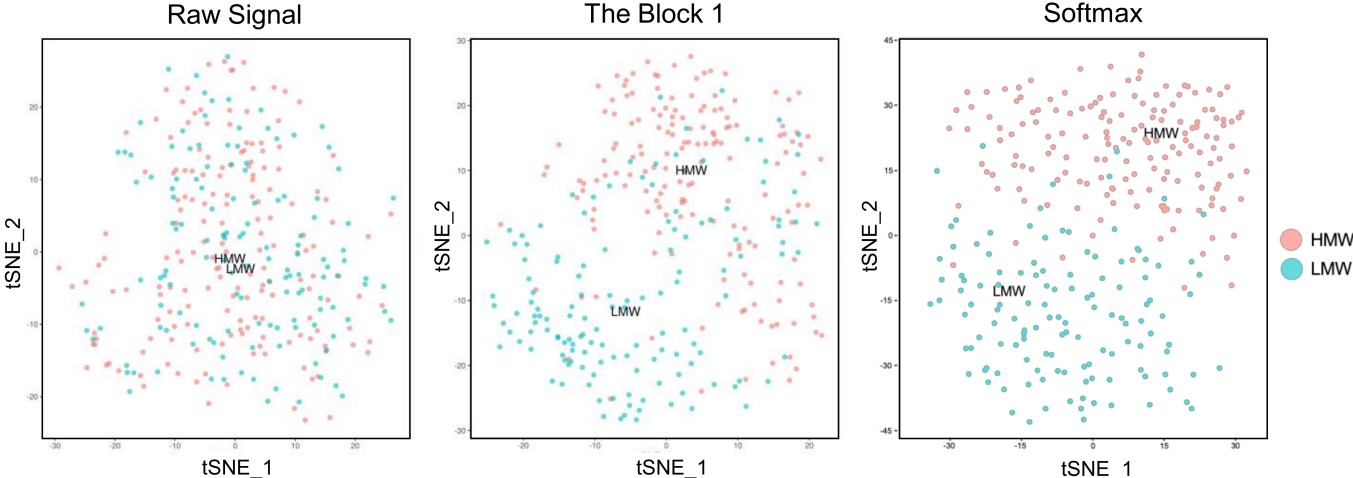

**Figure 8.** The t−SEN chart of the proposed method at three steps.

To present the classification ability and classification efficiency of the proposed model more visually, the confusion matrix of classification results generated by this method has been given in Figure 9. Furthermore, subject 10′s training is used as an example, and the ROC diagram is provided in Figure 10. In addition to this, loss and accuracy values are provided for the learning process of the training and validation sets based on the different iterations in Figure 11.

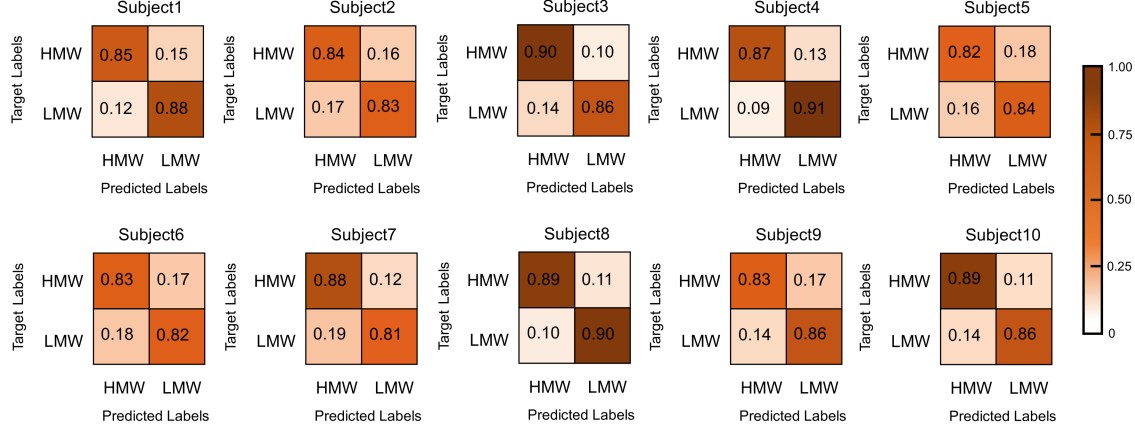

**Figure 9.** Confusion matrix of ten subjects.

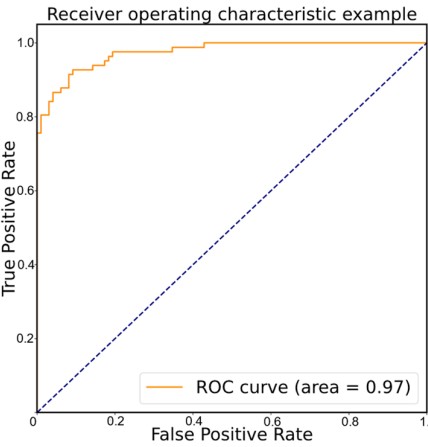

**Figure 10.** The ROC diagrams.

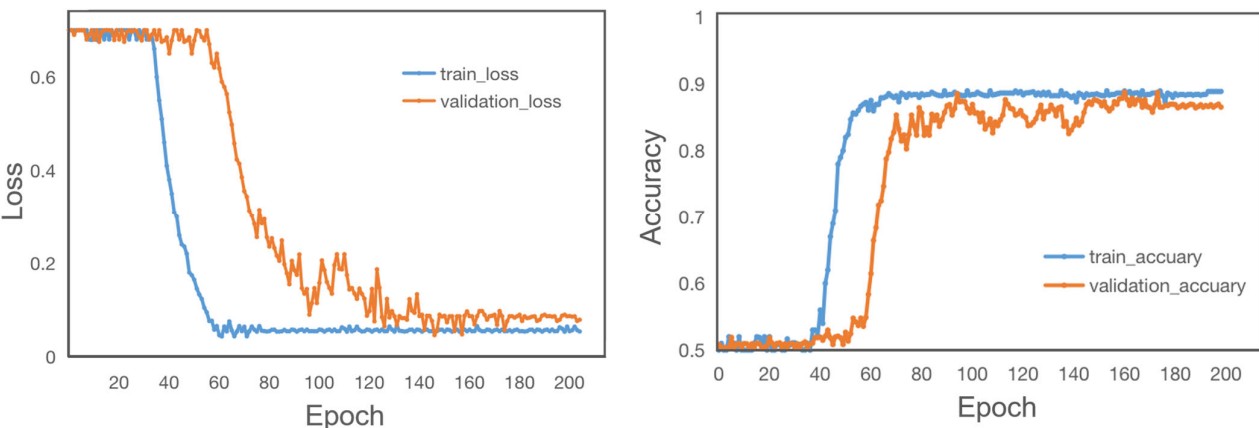

**Figure 11.** The loss and accuracy of the proposed method.

### 3.4. Ablation Experiment

In order to evaluate the classification ability of the model proposed in this paper, two comparison methods are proposed:

Method I: the model preprocessing stage is added to PCA to reduce the dimensionality, and then directly fed into the GCN model for training.

Method II: the GCN model is changed into a residual connection on the basis of method 1, and then the data are fed into the training; The last is the method proposed in this paper. The specific descriptions of the three methods are shown in Table 2, and the results of the three methods are shown in Table 3.

**Table 2.** Specific description of the three methods.

| Name | Description |
| --- | --- |
| Method I | GCN |
| Method II | GCN+Residual |
| Proposed | GCN+SE+Residual |

**Table 3.** Comparison of experimental accuracy of three methods for ten subjects.

| Subjects | Acc% | | |
|---|---|---|---|
| | Method I | Method II | Proposed |
| subject1 | 70.6 | 74.1 | 87.1 |
| subject2 | 68.5. | 74.6 | 83.4 |
| subject3 | 73.9 | 74.9 | 88.6 |
| subject4 | 74.9 | 78.3 | 89.2 |
| subject5 | 69.8 | 73.6 | 83.6 |
| subject6 | 71.2 | 73.9 | 83.4 |
| subject7 | 75.6 | 77.6 | 85.2 |
| subject8 | 74.9 | 78.4 | 89.7 |
| subject9 | 69.7 | 74.2 | 85.2 |
| subject10 | 72.8 | 75.7 | 87.2 |
| Average | 72.2 | 75.6 | 86.3 |

According to the data 9:1 division ratio, the data of ten subjects are sequentially used as the test set. The comparison experiments in Table 3 show that the residual connection of the proposed model has some improvement on the classification ability of the model and the accuracy is improved by about 3%. This phenomenon is explained by the structural properties of residual connections. It guarantees the stability of the network during training and ensures that the network does not experience gradient explosion. Additionally, in the residual connection, the model only needs to learn the number of residuals between different layers, so the model has a better optimization effect during training. The experimental results show that the accuracy of the model is directly improved by 15% after adding the SE block. This indicates that although the data belong to the traditional Euclidean space, training the model by the default ordering and then assigning the same weights causes information loss. Therefore, the primary and secondary relationships between the electrodes are considered to have a significant impact on the results.

*3.5. Classification Results Comparison between Several Methods*

In order to compare the classification ability of the proposed EEGCN models, eight different models are used to train the same dataset. The results are shown in Table 4. The first two of them are the methods proposed by Zheng et al. [21] and Qu et al. [20] based on machine learning. The rest of the six methods in the table are all based on deep learning. The three comparison methods, CNN, LSTM, and recurrent neural networks (RNN), all use the same structure and data as the network framework part in Figure 1. There is no residual connectivity and attention mechanism. The last three methods used for MW classification are deep learning methods that improve on the classical model. One classification method proposed by Pang et al. [17] is based on the stochastic configuration network (SCN). The subject-specific classifiers (SSCs) are built using the individual EEG data. Umer Asgher et al. [23] propose a method using SVM and CNN for MW classification, in which they select CNN with two convolutional layers, one max pool layer, and one fully connected layer before output. A deep hybrid model based on bidirectional long short-term memory (BLSTM) and LSTM has been proposed for the classification of workload levels by Debashis Das Chakladar et al. [24].

**Table 4.** Comparison of results of the three methods.

| Model | MAcc(%) | MKappa | MF1 |
|---|---|---|---|
| Transfer Learning [21] | 85 | 0.86 | 71.8 |
| CSSA [20] | 82 | 0.80 | 71.3 |
| CNN | 74.2 | 0.69 | 67.4 |
| LSTM | 71.8 | 0.74 | 69.6 |
| RNN | 70.3 | 0.77 | 69.2 |
| SCN+SSCs [17] | 75.91 | 0.65 | 68.2 |
| CNN+SVM [23] | 80.42 | 0.78 | 71.9 |
| BLSTM-LSTM [24] | 83.87 | 0.88 | 72.1 |
| EEGCN | 86.3 | 0.89 | 72.5 |

By comparing the results, we see that the classification results of EEGCN are significantly higher than others, especially the simple CNN, LSTM, and RNN networks. The comparison models perform poorly in classification, in addition to the fact that they do not fully extract the high-level feature of electrode signal correlation. This is also explainable according to the characteristics of the EEG signal. In contrast to the speech signal commonly used in natural language processing, the EEG signal is less volatile, with more interference signals. Therefore, just using a simple LSTM or RNN network in the network, with the exception of the inter-temporal features that can be extracted, the other capabilities are very weak. Nowadays, RGB pictures have rich features which are commonly used in image processing. The higher-level features can be continuously mined after continuous convolution. Constantly adding convolution layers in CNN can easily produce overfitting and gradient explosion. It can be observed that the simple stacked CNN model is not suitable for EEG signal processing. It can be shown experimentally that the classical deep learning models proposed in academia today are not fully applicable to this signal due to the stochastic non-smoothness of the EEG signal. The experimental results are not satisfactory when the data are inputted blindly. In the analytical processing of EEG signals, in order for the network to extract signal features from multiple perspectives, adjustments to the existing models are needed such as the three comparison methods that follow. Pang et al. [17] proposed the SCN method, which uses the same data as the proposed method. The results show that the range of SSC test accuracy is between 56.5% and 90.2% with an average of 75.9%. It can be seen that the classification ability of SCN is lower than the model proposed in this paper. In addition, we feed the dataset collected by the MATB-II platform into the method proposed by Umer Asgher et al. [23]. The data are processed by SVM and CNN, which select CNN with two convolutional layers, one max pool layer and one fully connected layer before output. The classification ability is not as good as the EEGCN model proposed in the paper, which has an average accuracy of 80.42 in the dataset collected in this paper. Finally, the dataset collected by the MATB-II platform is fed into the hybrid model proposed by Debashis Das Chakladar et al. [24]. The data are classified after BLSTM and LSTM, and the average accuracy reaches 83.87, which is slightly lower than the model proposed in this paper.

## 4. Discussion

To solve the problem of ignoring correlations in the MW classification task, we provide a new method. Firstly, the features are extracted in a hand-crafted way by PCA to obtain the raw EEG signal. Then, to feed the correlated features into the model, we firstly hand extract the mapping features. Then, they are fed into the GCN for learning. Later, PCA downscaling and attention mechanisms are added successively to speed up the training. Thereby, the classification ability of the model is improved, and the average accuracy reaches 86.3%. This combination of traditional dimensionality reduction methods and neural networks provides a new idea for future MW classification.

However, the training speed is still slow in the experiments, even though the number of parameters in the model is only 21M. The GCN used in the proposed model is a

transductive learning, which needs to load the training set and validation set at the same time. Additionally, each forward computation will count out all nodes, and the Laplacian matrix needs to be multiplied with the matrix composed of the features of all nodes during the convolution process. Thus, the space occupation of such a Laplacian matrix is N × N, so it is very memory- and video memory-intensive. Therefore, the hardware requirements are high in the application scenario. To continue solving this existential problem, it is envisioned whether the correlation between electrodes of different subjects is the same. If the assumption holds, the model can be optimized in combination with transfer learning methods. Thus, the amount of training required for the model can be reduced. How to speed up the training, reduce the memory consumption, and further improve the model accuracy is the next step of our work.

## 5. Conclusions

In this study, the correlation between electrodes is included. An SE block and residual connection-based GCN method is also proposed for MW classification. The model converts the raw EEG signal after PCA downscaling into a graph structure based on correlation coefficients between brain electrodes as input. It also uses the SE block and residual connection for training to obtain the final inference model.

Based on the analysis and discussion of the results, it can be concluded that:

(1) Extraction of graph features

The human brain has small-world properties. Correlations exist between each neuron. Therefore, we can view the human brain as a topological map structure. Further, the multiple one-dimensional independent signals collected by the EEG bubble that we fit to the shape of the human brain can also be abstractly represented as a graph composed of vertices and edges. After the pre-processing of the temporal-domain signals, such as through de-artificing, the correlation between each node is quantified using Pearson correlation coefficients. Thus, the inter-electrode graph features are extracted and subsequently fed into the GCN to model them from a multidimensional perspective.

(2) GCN network modifications

In the EEGCN model proposed in this paper, a PCA dimensionality reduction strategy is introduced in the preprocessing stage. An SE block is added to the GCN model to set learnable parameters for each dimension, so that the model is data-driven and mines the ratio of weights among different dimensions by itself. The HMW and LMW are classified in the EEG data collected by the MATB-II platform for the simulated flight task. An average accuracy of 86.3% is achieved in ten subjects. In comparative experiments using the same dataset, EEGCN shows a significant improvement over the traditional machine learning methods and classical deep learning models.

The method proposed in this paper can be applied in the future in some complex human–machine systems with high safety requirements, such as subway driving aircraft, piloting, manned spaceflight, and high-altitude operations. The real-time detection and evaluation of the operator's brain load can reasonably control the task demand and human brain load level in the human–machine system, which is very important for the system efficiency, safety, and human health [47].

**Author Contributions:** Conceptualization: H.Q.; methodology: H.Q.; software: Z.Z. (Zitong Zhao); validation: Z.Z. (Zitong Zhao) and H.Q.; formal analysis: Z.Z. (Zheng Zhang); investigation: Z.Z. (Zheng Zhang); resources: L.P.; data curation: C.L.; writing—original draft preparation: Z.Z. (Zitong Zhao); writing—review and editing: Z.Z. (Zitong Zhao) and H.Q.; visualization: L.P.; supervision: C.L.; project administration: C.L.; funding acquisition: H.Q. All authors have read and agreed to the published version of the manuscript.

**Funding:** This research is funded by National key research and development program of China (2020YFB1600704).

**Institutional Review Board Statement:** This study was conducted in accordance with the Declaration of Helsinki and approved by the Institutional Biological and Medical Ethics Committee, Beihang University (protocol code BM201900078, June 2019).

**Informed Consent Statement:** Informed consent was obtained from all subjects involved in the study.

**Data Availability Statement:** The raw data supporting the conclusions of this article are available on request from the corresponding author.

**Acknowledgments:** The authors are grateful to the subjects for their contributions to the experiment.

**Conflicts of Interest:** The authors declare no conflict of interest.

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
