# Peer review of "A Mental Workload Classification Method Based on GCN Modified by Squeeze-and-Excitation Residual"

_mathematics, doi:10.3390/math11051189_

Round 1

Reviewer 1 Report

The authors proposed a method for classifying mental workload based on a modified GCN. Although the problem studied is interesting, I have the following concerns that need to be addressed. Accordingly, the paper needs to be major revised.

1.       Recently, several papers have been published on mental workload classification using deep learning models. Many of them have provided significant results. How does your work differ from others?

2.       Many recent studies are not included in this study. It is necessary that more recent studies be presented. For example:

"Deep convolutional neural network for classification of sleep stages from single-channel EEG signals"

"Classification Mental Workload Levels from EEG Signals with 1D Convolutional Neural Network"

"Recognizing emotions evoked by music using CNN-LSTM networks on EEG signals"

"On time series cross-validation for deep learning classification model of mental workload levels based on EEG signals"

3.       The proposed method is sensitive to the values of its main control hyperparameters. How did you set the hyperparameters?

4.       Please provide evaluation criteria for the learning process for training and validation data based on different iterations.

5.       Please provide the confusion matrix and the t-SNE and ROC plots for the proposed model.

6.       Why did the authors use the subjects individually, and why did you not use all subjects together for training the model?

7.       Overall, some of the formulas need to be rewritten.

8.       Some of the symbols in the text are higher than the text (e.g., lines 183, 185, etc.). Please revise the manuscript.

Reviewer 2 Report

The paper "A mental workload classification method based on GCN modified by 2 squeeze-and-excitation residual" presents an interesting approach for mental workload classification using graph convolutional networks (GCN) and squeeze-and-excitation residual blocks. Overall, the paper is well-written and the method is clearly explained.  The experimental results are presented in a clear and informative manner, and the use of two public EEG datasets for evaluation adds to the validity of the results.

However, there are a few areas that could be improved:

1) The discussion of the limitations of the proposed method is limited. The authors should provide more insight into the limitations of the method.

2) The robustness of the method should be further evaluated. The authors could consider experimenting with different types of EEG signals or  different mental workload tasks to evaluate the robustness of the method.

3) The literature review could be more extensive. The authors should provide a more comprehensive overview of the current state-of-the-art in mental workload classification and how their work fits into the broader field.

4) The figures and tables should be more detailed, It would be better to have more detailed figures and tables that clearly show the results of the experiments.

5) Some computation complexity of this method should be added in the revised manuscript.

6) Based on your title, authors should discuss the future work in the conclusion section such as: Event‐triggered synchronization for stochastic delayed neural networks: Passivity and passification case.

7) Authors should check the language errors, typos, and etc througout the manuscript.

Reviewer 3 Report

A Squeeze-and-Excitation (SE) based Graph Convolution Network (GCN) has been proposed in this work to monitor the level of Mental Workload (MW) in complex labor production and human-machine interaction. First, the proposed method reduces the dimensionality of raw Electro-EncephaloGram (EEG) signals through Principal Component Analysis (PCA). Then, it transforms the reduced data into graph structure data, preserving the inter-neural association information. The GCN with the SE residual block is then utilized for the final classification of the mental workload levels.

In this proposed method, the SE block performs an adaptive recalibration of the channel-wise feature responses, while the residual connection enables the practical training of the network. Experiments conducted on raw EEG signals obtained from 10 healthy subjects demonstrate the efficacy and superiority of the proposed GCN-SE method compared to traditional models.

However, the authors are advised to ensure consistency in formatting and referencing throughout the paper. Additionally, the section names in the text should be updated to reflect their content accurately, and the figure captions should be more descriptive to provide sufficient information. Finally, it would also be beneficial for the authors to clarify if the compared models are state-of-the-art.

Round 2

Reviewer 1 Report

My comments have been addressed. the current manuscript is suggested for publication.

Reviewer 2 Report

The revised version has been well written, thus it can be suitable for publication.